# Effect of Critical Illness Insurance on Household Catastrophic Health Expenditure: The Latest Evidence from the National Health Service Survey in China

**DOI:** 10.3390/ijerph16245086

**Published:** 2019-12-13

**Authors:** Anqi Li, Yeliang Shi, Xue Yang, Zhonghua Wang

**Affiliations:** 1School of Health Policy Management, Nanjing Medical University, Nanjing 211166, China; lianqi_njmu@163.com (A.L.); 2004azure@163.com (Y.S.); njmuyangxue@163.com (X.Y.); 2Creative Health Policy Research Group, Nanjing Medical University, Nanjing 211166, China; 3Center for Global Health, Nanjing Medical University, Nanjing 211166, China

**Keywords:** critical illness insurance, household, catastrophic health expenditure, incidence and intensity

## Abstract

Background: China fully implemented the critical illness insurance (CII) program in 2016 to alleviate the economic burden of diseases and reduce catastrophic health expenditure (CHE). With an aging society, it is necessary to analyze the extent of CHE among Chinese households and explore the effect of CII and other associated factors on CHE. Methods: Data were derived from the Sixth National Health Service Survey (NHSS, 2018) in Jiangsu Province. The incidence and intensity of CHE were calculated with a sample of 3660 households in urban and rural areas in Jiangsu Province, China. Logistic regression and multiple linear regression models were used for estimating the effect of CII and related factors on CHE. Results: The proportion of households with no one insured by CII was 50.08% (1833). At each given threshold, from 20% to 60%, the incidence and intensity were higher in rural households than in urban ones. CII implementation reduced the incidence of CHE but increased the intensity of CHE. Meanwhile, the number of household members insured by CII did not affect CHE incidence but significantly decreased CHE intensity. Socioeconomic factors, such as marital status, education, employment, registered type of household head, household income and size, chronic disease status, and health service utilization, significantly affected household CHE. Conclusions: Policy effort should further focus on appropriate adjustments, such as dynamization of CII lists, medical cost control, increasing the CII coverage rate, and improving the reimbursement level to achieve the ultimate aim of using CII to protect Chinese households against financial risk caused by illness.

## 1. Introduction

In response to the call of the World Health Organization (WHO) and to promote population health status, many countries are exerting effort to achieve universal health coverage (UHC), which is a major Sustainable Development Goal. Achieving UHC means that all people receive essential health services without being exposed to financial hardship [1]. Since 2003, China has introduced and implemented a comprehensive package of basic medical insurance (BMI). The BMI system consists of urban employee basic medical insurance, urban resident medical insurance (URMI), new rural cooperative medical scheme (NRCMS), and the urban and rural medical assistance system. These basic medical insurance plans cover about 97% of the population (over 1.3 billion Chinese), which enabled China to achieve near UHC [2].

Despite nationwide coverage, the service provisions and financial protection of the Chinese BMI system are far from sufficient. The risk of various critical diseases (e.g., malignant tumor, acute myocardial infarction, coronary artery diseases, cerebral stroke, etc.) and their economic burden are dramatically increasing with an aging society. Critical illness caused 36.19 million deaths and 662.71 million years of life lost worldwide in 2016 [3]. The Global Burden of Disease Study 2017 also shows that non-communicable diseases accounted for 73.4% of deaths [4]. In China, mortality due to malignant tumor, cerebral vascular disease, heart disease, and respiratory system disease accounts for 80% of all deaths. Furthermore, high out-of-pocket costs for treating these critical illnesses can easily impoverish patients and their households. The economic burden of chronic diseases is as high as 2.6 trillion in China [5], and more than 40% of 72 million poor people re-entered poverty because of diseases in 2016 [6]. Healthcare expenses may overburden patients with critical illnesses, which are often financially devastating for their households. With a limited financial protection effect of the BMI system on reducing catastrophic health expenditure (CHE) in China [7,8,9], the economic burden of diseases in Chinese households is higher than generally appreciated, and will continue to increase as the population ages [3].

To protect urban and rural residents against large medical expenses and relieve medical expenditure burden, the China central government launched critical illness insurance (CII) in August 2012. It was piloted in more than 134 cities from 2013 and implemented nationally in 2016. CII, also called critical illness insurance for urban and rural residents, covers the enrollee of URMI and NRCMS. It reimburses individuals with critical illnesses for high out-of-pocket healthcare payments (OOP) exceeding a certain level after BMI reimbursement [10]. The central government proposes general guiding principles; for example, the total reimbursement rate is no less than 50% (adjusted to 60% in a 2019 government work report [11]) when medical bills for necessary treatments after reimbursement under the BMI system exceed annual per capita income levels. All provincial governments have made detailed rules about the financial mechanism, coverage, and reimbursement with respect to CII. This program does not require an additional premium from the insured; it is mainly funded through BMI surpluses. Therefore, it is an extension and supplement to the Chinese BMI system.

Total OOP equaling or exceeding 40% of the household capacity to pay (CTP) is considered catastrophic. This definition for CHE proposed by the WHO is widely accepted [3,12,13]. The CHE is an important indicator for domestic and foreign researchers to evaluate the effects of medical insurance systems. Many studies have assessed CHE worldwide. Barros et al. [14] estimated catastrophic healthcare expenditure in Brazil using different definitions (including >40% of household CTP), and the results indicated that health insurance coverage did not protect from CHE. A systematic review was undertaken by some researchers in Ghana to evaluate whether enrollment in the National Health Insurance Scheme reduces the likelihood of OOP and CHE. They adopted the thresholds of CHE that OOP for healthcare exceeds 20% of annual household income, 10% of household expenditures, and 40% of subsistence expenditures, respectively. Findings suggested that healthcare costs remain catastrophic for a large proportion of insured households in Ghana [15]. Two other studies by Xie et al. [16] and Guo et al. [17] evaluated the role of the Chinese new rural cooperative medical scheme (NRCMS) on protecting households from CHE using 40% of a household’s capacity to pay as the threshold. Both reported that CHE incidence and intensity decreased after NCMS reimbursement.

Alleviating the economic disease burden and protecting households from CHE were the most important targets of CII [10]. Although local governments have implemented CII for several years, it is far from achieving universal coverage. For example, approximately 50% of households in Jiangsu Province have not been insured by CII, according to the Sixth National Health Service Survey (NHSS) conducted in Jiangsu Province in 2018. Some studies have researched the implementation effect of CII in China. However, most focused on describing and comparing the design and development of the program policy [18,19,20,21] and do not provide strong empirical evidence on the topic. In recent years, several studies performed quantitative analyses. Fang et al. [22] evaluated the effect of CII on the share of medical expenses reimbursed and found that it increases the level of reimbursement to some extent. Zhao et al. [3] indicated that CII is only partially effective in financial risk protection under total medical expenses rather than covered medical expenses. Another study [23] focused on the impact of CII on utilization and costs of hospitalization before and after implementation and found that CII can promote inpatient service utilization but increases patients’ economic burden.

All individuals should have access to the health services they need without risk of financial ruin or impoverishment (WHO 2013) [24]. With China’s aging society, it is necessary to analyze the effect of CII on household CHE and economic burden of diseases. To the authors’ knowledge, due to a short period of time since broad CII implementation and the lack of empirical data, few studies have assessed the effect of CII on CHE. This study used data from the Sixth National Health Service Survey (2018) to calculate the extent of catastrophic health expenditure in urban and rural households in Jiangsu Province and evaluated the effect of critical illness insurance and associated factors on household catastrophic health expenditure. This study clarifies the implementation effect of its critical illness insurance on financial risk protection in Jiangsu Province; it also provides some significant empirical evidence for reforming China’s healthcare insurance.

## 2. Materials and Methods

### 2.1. Data Source

The primary data used in this study were derived from the Sixth NHSS (2018) in Jiangsu Province. The NHSS has been organized by the Chinese National Health Commission every 5 years since 1993, and the Provincial Commission of Health is responsible for the specific survey in their region. The survey aims to understand resident health status and needs for health services in urban and rural China, so as to understand the allocation and utilization of health resources and other information. The sixth NHSS was conducted in 31 provinces in September 2018. According to economical and effective sampling criteria, it used a multistage and stratified cluster sampling method based on administrative region division. With the help of the Center for Health Statistical and Information of National Health Commission, 156 county-level units (counties or urban districts) were sampled, and within each county-level unit, 5 villages and community units were selected as the primary sampling units. Finally, 60 households were selected within every administrative village and community unit using systematic random sampling. The respondents of the survey are the resident population among the sampled households. The resident population refers to all registered and non-registered people who have lived in the household within the past six months. It also includes babies born less than six months old, newly-married spouses, old people supported in turn, and primary and middle school students, but does not include non-family members, such as a nanny. Jiangsu Province is located in the Yangtze River Delta, eastern coast of Mainland China with a population of 80.29 million, 68.76% and 31.24% of the residents lived in urban and rural areas, respectively. At the end of 2017, the per capita Gross Regional Product (GRP) of Jiangsu Province was CNY 107150, which was ranked fourth among 31 provinces. The NHSS of Jiangsu Province was conducted in September 2018. Nineteen county-level units were sampled. A number of 11,550 respondents were included. After the elimination of missing individual value, the final household sample size was 3660.

The questionnaire contents include demographic and socioeconomic characteristics, healthcare services need and utilization, and medical insurances [25]. The questionnaire is structured and has good validity and reliability. Supervisors guided and inspected every step of the survey for quality control. Investigators were trained to conduct qualified face-to-face interviews. District survey managers checked the questionnaires at the end of each day to avoid missing information or logic errors. Moreover, 5% of all sampled households were randomly selected for revisits to check data accuracy [26]. This study was on a household basis, and most variables were defined in terms of household unit. However, several variables (registered type, marital status, education, employment status) were the characteristics of the head of household. For evaluation accuracy, the variable of BMI was not included in the model because of nearly full coverage in China. Detailed descriptions of variables are shown in Table 1.

Among all households included, 54.04% (1978) were registered as urban and 45.96% (1682) as rural. The proportion of households that no one insured by CII is 50.08% (1833), 72.65% of which were urban households. Regarding demographic classification of household head, 87.27% were married, 70.03% had less than lower secondary education, and nearly 30% were unemployed or retired.

### 2.2. Data Analysis

#### 2.2.1. Definition

With regard to the definition of CHE, we followed the one proposed by WHO: OOP is considered as catastrophic if it equals or exceeds 40% of the household’s CTP [27]. CTP is known as non-food household expenditure. Some studies adopt another definition that CHE occurs if OOP comprise ≥10% of total household expenditures. We used the former because subtracting food expenses can partly avoid deviation in measurement that is often ignored in poor households that cannot afford catastrophic payments [28].

#### 2.2.2. Measuring CHE Incidence and Intensity

*E* is an indicator determining whether CHE occurred in a household, which is represented as a binary variable:(1)$$E_{i}=\left\{\begin{array}{c} 0\\ 1 \end{array}\begin{array}{c} if\\ if \end{array}\begin{array}{c} T_{i}/\left(x_{i}-f\left(x\right)\right)<z\\ T_{i}/\left(x_{i}-f\left(x\right)\right)\geq z \end{array}\right\}$$
where *T* is the OOP payment for healthcare by household *i*, *x* is the total expenditure of the household, *f*(*x*) is the food expenditure of a household, and z is the given catastrophic threshold.

CHE is usually assessed in terms of incidence and intensity. Headcount (*HC*) is used to measure the incidence of CHE, while overshoot (*O*) and mean positive overshoot (*MPO*) are used to measure intensity. HC indicates the percentage of households whose OOP health expenditure, as a fraction of their total non-food expenditures, exceeds the specified threshold (*z*). When N represents the sample size, HC is estimated as follows:(2)$$\mathrm{HC}=\frac{1}{N}\overset{N}{\underset{i=1}{\sum }}E_{i}$$

The catastrophic payment overshoot (*O*) describes the average extent to which OOP health expenditure exceeds the chosen threshold of the entire sample. That is to say, it reveals the average CHE severity for all households. *O* and *O_i_* (the household overshoot) are estimated as:(3)$$O=\frac{1}{N}\overset{N}{\underset{i=1}{\sum }}O_{i}$$
(4)$$O_{i}=E_{i}\left(\left(T_{i}/\left(x_{i}-f\left(x\right)\right)\right)-z\right)$$

The MPO measures the mean overshoot among households with CHE at the given threshold in the sample, which can be calculated as follows:(5)$$\mathrm{MPO}=\frac{O}{HC}$$

#### 2.2.3. Regression Analysis on Associated Factors of CHE

Since incidence (*E_i_*) is a binary variable, logistic regression modeling was applied to estimate the effects of CII and related factors on the incidence of CHE. In addition, multiple linear regressions were used to analyze the associated factors of CHE intensity since intensity (*O_i_*) is a continuous variable. Assuming *x_k_* is factors related to CHE, and *Y* indicates the incidence or intensity, the model is shown below:(6)$$Y_{i}=\alpha +\sum_{k}\beta_{k}x_{ki}+\varepsilon_{i}$$

All statistical analysis was performed with Stata software version 14.0 (Stata Corp, College Station, TX, USA), and *p* < 0.1 was established as the level of statistical significance. In addition, we calculated the incidence and intensity of CHE of urban and rural households, respectively, based on five thresholds: 20%, 30%, 40%, 50%, and 60% of CTP, which was done to observe the changes of incidence and intensity at different levels. When we analyzed the determinants of CHE, we used the 40% threshold in accordance with the WHO definition of CHE.

## 3. Results

### 3.1. Characteristics of Urban and Rural Households

Table 2 shows the description statistics of characteristics of rural and urban households. Among all households included, 54.04% (1978) were registered as urban and 45.96% (1682) as rural. The proportion of households with one or more members insured by CII was 49.92%, and the average number of household members having CII was 1.44. Most (83.06%) household heads registered as rural reported having a less than lower secondary education, which was much higher in comparison with household heads registered as urban (58.95%). There were more tertiary educated subjects in urban areas (17.80%) compared to rural (2.56%). Employed household heads in rural areas accounted for 72.59%, which was nearly tripled the proportion of retired (7.79%) and unemployed (19.62%). In urban households, almost half of householders were retired or unemployed. The mean household size in rural areas (3.36) is higher than that in urban areas (2.98). It is noted that 76.46% of rural households had one or more members (2.40 on average) with CII, while only 27.35% of households in urban areas included members covered by CII (0.63 on average). Overall, 22.90% of sampled households reported having members with ≥2 chronic diseases, and there was no significant difference between urban (23.36%) and rural (22.35%) households. The percentages of households having one or more members ≥65 years of age were 34.36% and 41.30% in rural and urban households, respectively. With regard to healthcare utilization, the proportion of households with members having sought outpatient or inpatient care was significantly higher in rural area (31.45% and 29.31%) in comparison with the urban area (25.78% and 26.49%) Furthermore, the proportions of households with one or more member having sought outpatient and inpatient care from a second-level hospital or above were 14.4% and 25.08%, respectively.

### 3.2. CHE Incidence and Intensity in Urban and Rural Households

Table 3 presents the incidence (headcount) and intensity (overshoot) of CHE. CHE incidence and intensity in sampled households dropped remarkably as the threshold level increased from 20% to 60%. We observed significantly higher headcount and overshoot at any given threshold in rural households compared with urban households. At the lowest 20% threshold, households incurring CHE demonstrated the highest incidence (51.31%), overshoot (16.25%), and MPO (31.67%) among rural households. One in five (20.10%) rural households experienced catastrophe even at the highest 60% threshold. Using a widely used threshold of 40%, CHE incidence rates among urban and rural households were 26.54% and 32.28%, respectively, and the intensity indicates that urban and rural households on average spent 6.46% (overshoot) and 8.44% (overshoot), respectively, over the 40% catastrophic threshold. The MPO for households facing CHE was estimated as 25.25%. The average OOP was 65.25% (40% + 25.25%) of their total non-food expenditure. The corresponding value was slightly higher in rural (26.15%) than in urban (24.32%) households.

### 3.3. Associated Factors of CHE Incidence

The associated factors of the CHE incidence (at a non-food threshold of 40%) are shown in Table 4. Households with any members insured by CII were significantly less likely to experience CHE compared with households having no members insured by CII, but the number of household members with CII did not have any significant effect on CHE incidence. Considering some potential interaction effect, we include an interaction variable of the number of household members having CII and household members >65 years old in the regression model. It showed that in households with >65-year-old members, more household members having CII will increase the incidence of CHE significantly (OR = 1.077, 95%CI = (0.986, 1.175)). In addition, household heads with an education level of tertiary were significantly less likely to incur CHE compared to those with a low education level (odds ratio (OR) = 0.723, 95%CI = (0.521, 1.003)). The results indicate a significant correlation between CHE incidence and household size (OR = 0.933, 95%CI = (0.867, 1.004)), which means that CHE are more likely to occur in small, elderly households. High household income groups had a lower likelihood of CHE. In addition, households with members having ≥2 chronic diseases demonstrated a significantly higher likelihood of CHE (OR = 2.040, 95%CI = (1.694, 2.458)). Healthcare utilization of household members was negatively associated with CHE incidence. Having one or more members that visited hospitals for outpatient or inpatient services significantly increased the likelihood of CHE (OR = 1.421; 1.736, 95%CI = (1.154, 1.750); (1.114, 2.704)). Households with members having ever visited second-level hospitals or above for outpatient or inpatient services had a higher CHE incidence (OR = 1.270; 2.111, 95%CI = (0.973, 1.656); (1.332, 3.344)).

### 3.4. Associated Factors of CHE Intensity

Table 5 shows the multiple linear regression results for CHE intensity. We found a mixed impact of the CII status of household on CHE intensity. Having one or more household members with CII was positively associated with CHE intensity compared with having no CII-covered household members. However, as the number of household members insured by CII increased, the intensity of CHE significantly decreased. The results also revealed a statistical correlation between marital and education status of household heads and CHE intensity. Unmarried or individuals having a medium-level education (upper secondary and vocational training) were positively associated with CHE intensity. Household head employment status was also significantly correlated with CHE intensity. Specifically, retired or unemployed status significantly increased CHE intensity in comparison with the employed. Although living in rural areas was associated with a lower likelihood of CHE in Table 4, the intensity of CHE was significantly higher in rural households compared to urban households. The presence of household members >65 years old was a significant contributor to CHE intensity. The interaction variable of household members >65 years old and number of household members having CII was negatively associated with CHE intensity. In addition, the association between household size and CHE intensity was significantly positive. Household income levels negatively associated with CHE intensity, and this effect was estimated to be greater among high-income groups. Households with any member with at least two chronic diseases significantly increased CHE intensity.

Healthcare utilization of household members was positively correlated with CHE intensity, which means households with any member that sought outpatient or inpatient services had a significantly higher intensity of CHE. Furthermore, CHE intensity significantly increased when household members had gone to the second-level hospitals or above for inpatient treatment. However, there was no association with CHE intensity when members in a household sought outpatient services in high-level hospitals.

## 4. Discussion

Implementing CII is an important step in the Chinese government’s efforts to reduce CHE and alleviate household economic burden due to illness. As China’s population rapidly ages, more residents are suffering from serious diseases, and households are faced with heavy financial burdens. Following nationwide CII implementation, it is essential to study the effect on household financial risks of diseases. Our study used the latest data to calculate the incidence and extent of CHE among rural and urban households and estimate the effect of CII and other factors on CHE incidence and intensity.

Among the 3660 included households in Jiangsu Province, the proportion with any members insured by CII in rural areas was higher than that in urban areas (76.46% vs. 27.35%), which demonstrates the inadequate and imbalanced coverage of CII. The proportion of all sampled households facing CHE at OOP ≥40% of non-food expenditure was 29.18%, which was greater than the proportion (16.5%) reported in another study based on 2015 national data [29]. In Table 3, The CHE incidence (Headcount) and intensity (Overshoot) of rural households were significantly higher than those of urban households, irrespective of the threshold used. This result is similar to a study conducted in three cities in eastern China [30]. Moreover, we found the CHE of rural households was >66.15% of their non-subsistence expenditure (MPO). These findings in regard to CHE incidence (Headcount) and intensity (Overshoot) of households in rural and urban areas indicate that households, especially in rural areas, are still vulnerable to high medical care costs. Lower income level and the absence of social security in rural areas are among the possible causes. Thus, the government should implement more measures focused on rural households to improve the social security system and reduce their CHE.

Regarding the associated factors of CHE in Jiangsu Province, we found that households with a low education householder, lower household income, members with ≥2 chronic diseases, and utilizing outpatient and inpatient services have a higher likelihood of CHE incidence, which is in accordance with other studies [3,31,32,33,34]. The above factors are also associated with a higher intensity of CHE. The coefficient of the interaction variable of household members >65 years old and number of household members having CII was −0.006 (*p* < 0.05), which indicates that, compared to households without members >65 years old, the number of members covered by CII in households with members >65 years old have stronger negative effect on the intensity of CHE. With regard to CHE incidence, the effect of the interaction variable was significantly positive, which means that households with >65-year-old members are more likely to have CHE when the number of CII-insured members in the household increases. The CII seems to provide protection for households having members >65 years old who are also the insured of CII from high intensity of CHE but fail to reduce the occurrence of CHE among households. We supposed that more CII patients in a household might have a higher likelihood of having expensive diseases in that household. Also, most of the members insured by CII in a household may be the elderly. They are easily suffered from diseases and are vulnerable to higher healthcare expenses. Therefore, it is necessary to further focus policy efforts on elderly and impoverished households, such as increasing welfare subsidies or extending medical assistance programs to enhance financial protection and reduce CHE. Previous reports described significant associations between the type of healthcare facility and CHE [28,35]. Our study also found that seeking inpatient services in second-level hospitals or above increased CHE incidence and intensity, which suggests that higher medical costs of high-level hospitals and hospitalization may result in more catastrophic health expenses. In addition, chronic diseases cause both suffering to patients and catastrophic financial burden to households, as demonstrated in several studies [29,31,32,33,34,36]. Therefore, increasing the medical insurance level of patients with chronic diseases and improving chronic disease management could reduce household CHE and provide protection against medical economic risks.

Rural registration had no significant association with CHE incidence, contrary to some existing studies [29,30], but had significantly higher CHE intensity compared with urban households. A possible explanation is that there may be non-use of needed healthcare services by patients in rural households because of low income [37]. However, rural patients utilizing health services tend to go to more expensive high-level hospitals because they offer advanced medical equipment and technology compared to first-level hospitals. Therefore, OOP of rural households may far exceed their non-food expenditure [38]. The government should take the reimbursement level of rural residents and improvement of first-level hospital services into account. Household size means the number of persons living together in one household. The result of our study that household size was negatively associated with the likelihood of CHE is in agreement with previous reports [29,39,40]. However, several published studies pointed out that large household size is significantly associated with high CHE [35,41]. We found a slightly positive correlation between household size and CHE intensity. These findings indicate that larger families protect against CHE incidence but not intensity. As the characteristic result in Table 1 indicated, rural household size is significantly higher than urban. Larger families are usually found in Chinese rural areas. In China, a nuclear family, consisting of a married couple and their unmarried children, is the mainstream form of family structure in urban areas. While in rural areas, there are more households with three or four generations living together, which are called a stem family or joint family. And generally, the proportion of the elderly in rural households is higher than that in urban areas because the majority of the elderly are employed in agriculture, and they are used to living in rural areas. The young and middle-aged tend to go to cities for better employment and live there. Although larger families may have a good economic ability to reduce the risk of CHE, they are more likely to include more elderly members or children who are susceptible to illness and incur higher costs [30].

Previous studies concerning the effect of CII on relieving financial burden have yielded mixed results. Some studies held the view that the effect is limited from the perspective of CII reimbursement level [22,23,42], while other studies indicated that economic burden decreased after CII implementation [3,43,44]. In our study, a household with members having CII was a factor to protect households from CHE incidence, but it significantly increased CHE intensity. The number of household members insured by CII did not affect the incidence at a significant level, but this factor significantly decreased the intensity of CHE. A possible explanation is that having CII actually improves households’ capacity to resist economic risk due to critical diseases, and therefore, significantly decreased CHE incidence. However, CII implementation also induced residents’ demands for medical services [23], which could lead to higher healthcare expenses, even including uncovered medical expenses that are not on the CII list. Therefore, once CHE occurs in a household, the intensity will be greater, especially in lower-income households. As the number of the insured by CII in a household increased, the association of CII with incidence became insignificant. Instead, households with more CII-covered members decreased CHE intensity.

The main purpose of CII is to provide protection against CHE; however, our study showed that this objective has not been fully achieved in Jiangsu Province. In view of the possible causes listed above, policy efforts should focus on the following aspects. First, strategies should be developed to promote the inclusion of more households in CII to effectively decrease household CHE incidence. At the same time, it is necessary for the government to establish a dynamic reimbursement list of CII to expand coverage and reduce uncovered medical expenses. Second, due to the significant effect of the number of CII-covered household members on CHE intensity, the CII coverage rate in Jiangsu province should be continually increased to better relieve economic burden associated with disease. Third, to better exert the protective effect of CII, some medical cost control mechanisms should be established to reduce irrational increases of medical expenses exceeding actual needs. In addition, rural households are in urgent need of policy support because of the higher incidence and intensity of CHE, lower-income, and less social security. Therefore, the government should properly increase the reimbursement rate or reduce the deductible line of CII in rural areas to further alleviate household economic burden of diseases.

Some limitations of this study must be noted here. (1) We calculated household medical expenses, excluding indirect expenses (e.g., transport, food, accommodation costs, etc.) for patients and companions, which is reported to account for a fair proportion of total OOP. This conclusion may lead to an underestimation of CHE. (2) Due to limited access to data, our study used a sub-sample of NHSS (2018) data from Jiangsu Province, which may be not nationally representative. However, Jiangsu Province is a relatively developed area and is typical in the implementation of CII among 31 provinces surveyed. (3) We evaluated the effect of CII on CHE by measuring the incidence and intensity; we did not focus on specific OOP and the actual reimbursement ratio because of a lack of data. Further studies will be done in the future.

## 5. Conclusions

To the best of our knowledge, some theoretical studies researching the implementation effect of CII in China mainly focused on describing and comparing the design and development of the program policy. Due to a short period of time since broad CII implementation and lack of empirical data, few existing studies provide strong empirical evidence on the effect of CII on CHE. Our study, basing on Chinese latest and authoritative data of the representative region, not only reveals urban–rural discrepancies in the incidence and intensity of CHE but also show the mixed effect of CII and some other associated factors on CHE. More government measures focusing on rural households should be implemented to reduce their catastrophic medical expenses. Policy adjustments such as a dynamic CII list, medical cost control mechanisms, broader CII coverage, or improved reimbursement levels should be developed to provide solid protection for Chinese households against financial medical risk. Additionally, other socioeconomic factors significantly affecting CHE indicate that policies aimed at reducing CHE should address the socioeconomic factors of healthcare outcomes among urban and rural households. Our study clarifies the implementation effect of its CII on financial risk protection in Jiangsu Province, and it also provides reference and guidance for reforming China’s healthcare insurance.

## Figures and Tables

**Table 1 ijerph-16-05086-t001:** Description of explanatory variables.

Variable	Description	Survey Questions
**Household Head Variables**		
Registered Type	=1 if registered as rural; =0 if registered as urban	Question: What’s your registered type?
Education Status	Less than lower secondary/upper secondary vocational training/tertiary	Question: What’s your education level?
Employment Status	Employed/retired/unemployed	Question: What’s your employment status?
Marital Status	=1 ifmarried; =0 if single, widowed, divorced, or other	Question: What’s your marital status?
**Household Variables**		
Household Size	Number of persons living together in one house	Number of household members
Household CII Status	=1 if one or more members have CII; =0 otherwise	Question: Are you insured by the Critical Illness Insurance?
Number of Household Members Having CII		Number of household members insured by CII
Household Income Level	≤30,000/30,000–50000/50,000–78,500/78,500–100,000/<100,000	Total household income is grouped by quintile. Question: What was your total household income in the previous year (2017)?
Household Members with ≥2 Chronic Illness	=1 if anyone in a household has two or more chronic illnesses; =0 otherwise	Questions: Do you have confirmed hypertension? Do you have confirmed diabetes? Do you have other chronic diseases diagnosed?
Household Members>65 Years Old	=1 if one or more members >65 years old; =0 otherwise	Question: Year of birth.
Outpatient Utilization Status of Household Members	=1 if any household member visited a doctor because of any illness during the past 2 weeks; =0 otherwise	Question: Have you seen a doctor because of feeling ill during the two weeks prior to the survey?
Inpatient Utilization Status of Household Members	=1 if any household member hospitalized during the year before the survey; =0 otherwise	Question: In the year before the investigation, did you live in a hospital because of illness, physical examination, delivery, etc?
Outpatient Service in High-Level Hospital	=1 if anyone in a household went to a second-level hospital or above for outpatient treatment; =0 otherwise	Question: where was your first visit in the two weeks before the survey?
Inpatient Service in High-Level Hospital	=1 if anyone in a household went to a second-level hospital or above for inpatient treatment; =0 otherwise	Question: Which of the following medical and health institution are you hospitalized this time?

Note: CII, critical illness insurance.

**Table 2 ijerph-16-05086-t002:** Characteristics of rural and urban households in Jiangsu Province.

Variable	Total *N* = 3660	Rural Household *N* = 1682	Urban Household *N* = 1978	*p*-Value
Marital Status of Household Head				0.417
Married	87.27% (3194)	87.75% (1476)	86.86% (1718)	
Unmarried (Single, Widowed, Divorced, or Other)	12.73% (466)	12.25% (206)	13.14% (260)	
Education Status of Household Head				0.000
Less than Lower Secondary	70.03% (2563)	83.06% (1397)	58.95% (1166)	
Upper Secondary Vocational Training	19.18% (702)	14.39% (242)	23.26% (460)	
Tertiary	10.79% (395)	2.56% (43)	17.80% (352)	
Employment Status of Household Head				0.000
Employed	60.49% (2214)	72.59% (1221)	50.20% (993)	
Retired	23.14% (847)	7.79% (131)	36.20% (716)	
Unemployed	16.37% (599)	19.62% (330)	13.60% (269)	
Household Size (Mean)	3.16	3.36	2.98	0.000
Household CII Status				0.000
No One Insured	50.08% (1833)	23.54% (396)	72.65% (1437)	
One or More Members Insured by CII	49.92% (1827)	76.46% (1286)	27.35% (541)	
Number of Household Members Having CII (Mean)	1.44	2.40	0.63	0.000
Household Income Level				0.000
≤¥30,000	24.23% (887)	36.09% (607)	14.16% (280)	
¥30,000–50,000	19.86% (727)	23.42% (394)	16.84% (333)	
¥50,000–78,500	15.90% (582)	13.20% (222)	18.20% (360)	
¥78,500–100,000	21.07% (771)	15.93% (268)	25.43% (503)	
>¥100,000	18.93% (693)	11.36% (191)	25.38% (502)	
Household Members with ≥2 Chronic Diseases				0.472
All Household Members Have One or No Chronic Disease	77.10% (2822)	77.65% (1306)	76.64% (1516)	
Any Member with ≥2 Chronic Diseases	22.90% (838)	22.35% (376)	23.36% (462)	
Household Members >65 Years Old				0.000
No One >65 Years Old	61.89% (2265)	65.64% (1104)	58.70% (1161)	
One or More Members >65 Years Old	38.11% (1395)	34.36% (578)	41.30% (817)	
Outpatient Utilization Status				0.000
No One Utilized	71.61% (2621)	68.55% (1153)	74.22% (1468)	
One or More Member Utilized	28.39% (1039)	31.45% (529)	25.78% (510)	
Inpatient Utilization Status				0.058
No One Utilized	72.21% (2643)	70.69% (1189)	73.51% (1454)	
One or More Member Utilized	27.79% (1017)	29.31% (493)	26.49% (524)	
Outpatient Service in A High-Level Hospital				0.104
No One Went to A Second-Level Hospital or Above for Outpatient Treatment	85.60% (3133)	86.62% (1457)	84.73% (1676)	
Anyone Went to A Second-Level Hospital or Above for Outpatient Treatment	14.40% (527)	13.38% (225)	15.27% (302)	
Inpatient Service in High-Level Hospital				0.547
No One Went to A Second-Level Hospital or Above for Inpatient Treatment	74.92% (2742)	75.39% (1268)	74.52% (1474)	
Anyone Went to A Second-Level Hospital or Above for Inpatient Treatment	25.08% (918)	24.61% (414)	25.48% (504)	

Note: CII, critical illness insurance.

**Table 3 ijerph-16-05086-t003:** Incidence and intensity of catastrophic health expenditure (CHE_ in urban and rural households in Jiangsu Province.

		Catastrophic Threshold (Share of Non-Food Household Expenditure)				
		20%	30%	40%	50%	60%
All Households	Headcount	48.09%	36.15%	29.18%	23.74%	17.40%
	Overshoot	14.53%	10.52%	7.37%	4.87%	3.04%
	Mean Positive Overshoot	30.22%	29.10%	25.25%	20.53%	17.45%
Urban Households	Headcount	45.35%	33.27%	26.54%	21.03%	15.12%
	Overshoot	13.07%	9.33%	6.46%	4.23%	2.63%
	Mean Positive Overshoot	28.82%	28.04%	24.32%	20.11%	17.42%
Rural Households	Headcount	51.31%	39.54%	32.28%	26.93%	20.10%
	Overshoot	16.25%	11.92%	8.44%	5.63%	3.51%
	Mean Positive Overshoot	31.67%	30.15%	26.15%	20.91%	17.48%
Difference	Headcount	0.060 ***	0.063 ***	0.057 ***	0.059 ***	0.050 ***
	Sd	0.017	0.016	0.015	0.014	0.013
	Overshoot	0.032 ***	0.026 ***	0.020 ***	0.014 ***	0.008 ***
	Sd	0.007	0.006	0.005	0.004	0.003

Note: Difference is the difference of incidence and intensity between rural and urban households; *** significant at 1%. SD, standard deviation.

**Table 4 ijerph-16-05086-t004:** Logistic regression on the incidence of CHE (at a non-food expenditure threshold of 40%).

Variables	Incidence of CHE		
	OR	SD	95% CI
Marital Status of Household Head (Ref: Single, Widowed, Divorced or Other)	0.948	0.102	(0.768,1.170)
Education Status of Household Head (Ref: Less than Lower Secondary)			
Upper Secondary Vocational Training	0.831	0.092	(0.669,1.032)
Tertiary	0.723 *	0.121	(0.521,1.003)
Employment Status of Household Head (Ref: Employed)			
Retired	1.107	0.126	(0.886,1.383)
Unemployed	1.068	0.121	(0.856,1.333)
Registered Type of Household Head (Ref: Urban)	0.775	0.077	(0.638,1.103)
Household Members>65 Years Old (Ref: None)	1.138	0.126	(0.916,1.414)
Household Size	0.933 *	0.035	(0.867,1.004)
Household Income Level (Ref: ≤¥30,000)			
¥30,000–50,000	0.315 ***	0.036	(0.252,0.395)
¥50,000–78,500	0.254***	0.034	(0.196,0.330)
¥78,500–100,000	0.186 ***	0.025	(0.142,0.243)
>¥100,000	0.091 ***	0.015	(0.065,0.127)
Household Members with ≥2 Chronic Illness (Ref: None)	2.040 ***	0.194	(1.694,2.458)
Outpatient Utilization of Household Members (Ref: None)	1.421 ***	0.151	(1.154,1.750)
Inpatient Utilization of Household Members (Ref: None)	1.736 **	0.393	(1.114,2.704)
Household CII Status (Ref: No One Has CII)	0.760 *	0.109	(0.574,1.006)
Number of Household Members Having CII	1.026	0.051	(0.930,1.132)
Outpatient Service in High-Level Hospital (Ref: None)	1.270 *	0.172	(0.973,1.656)
Inpatient Service in High-Level Hospital (Ref: None)	2.111 ***	0.496	(1.332,3.344)
Household Members >65 Years Old × Number of Household Members Having CII (Ref: Number Of Members Having CII in A Household without Members over 65)	1.077 *	0.048	(0.986,1.175)

Note: * significant at 10%, ** significant at 5%, *** significant at 1%. CHE, catastrophic health expenditure; CI, confidence interval; CII, critical illness insurance; OR, odds ratio; SD, standard deviation.

**Table 5 ijerph-16-05086-t005:** Multiple linear regression on the intensity of CHE (at a non-food expenditure threshold of 40%).

Variables	Intensity of CHE		
	Coefficient	SD	95% CI
Marital Status of Household Head (Ref: Single, Widowed, Divorced, or Other)	0.046 ***	0.006	(0.034,0.058)
Education Status of Household Head (Ref: Less than Lower Secondary)			
Upper Secondary Vocational Training	0.011 *	0.006	(−0.001,0.023)
Tertiary	0.009	0.008	(−0.007,0.025)
Employment Status of Household Head (Ref: Employed)			
Retired	0.019 ***	0.006	(0.007,0.032)
Unemployed	0.034 ***	0.007	(0.021,0.047)
Registered Type of Household Head (Ref: Urban)	0.009 *	0.005	(−0.001,0.020)
Household Members >65 Years Old (Ref: None)	0.045 ***	0.006	(0.033,0.058)
Household Size	0.005 **	0.002	(0.001,0.009)
Household Income Level (Ref: ≤¥30,000)			
¥30,000–50,000	−0.052 ***	0.007	(−0.065,0.039)
¥50,000–78,500	−0.068 ***	0.008	(−0.083,−0.053)
¥78,500–100,000	−0.080 ***	0.007	(−0.095,−0.066)
>¥100,000	−0.107 ***	0.008	(−0.124,−0.091)
Household Members with ≥2 Chronic Illness (Ref: None)	0.041 ***	0.006	(0.030,0.052)
Outpatient Utilization of Household Member (Ref: None)	0.019 ***	0.006	(0.006,0.031)
Inpatient Utilization of Household Member (Ref: None)	0.049 ***	0.014	(0.022,0.077)
Household CII Status (Ref: No One Has CII)	0.025 ***	0.008	(0.010,0.041)
Number of Household Members Having CII	−0.005 **	0.003	(−0.010,−0.000)
Outpatient Service in High-Level Hospital (Ref: None)	0.003	0.008	(−0.012,0.019)
Inpatient Service in High-Level Hospital (Ref: None)	0.025 *	0.014	(−0.003,0.054)
Household Members >65 Years Old × Number of Household Members Having CII (Ref: Number of Members Having CII In A Household without Members Over 65)	−0.006 **	0.003	(−0.011,−0.001)

Note: * significant at 10%, ** significant at5%, *** significant at 1%. CHE, catastrophic health expenditure; CI, confidence interval; CII, critical illness insurance; SD, standard deviation.

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
