# Peer review of "Effect of Critical Illness Insurance on Household Catastrophic Health Expenditure: The Latest Evidence from the National Health Service Survey in China"

_ijerph, 2019, doi:10.3390/ijerph16245086_

Round 1

Reviewer 1 Report

Overall, I found the study to be well-designed and the methods were described clearly, for the most part. Please find my specific comments below:

Introduction

Overall, this section nicely frames the study topic; however, I find that there are too many acronyms to keep track off. In addition, please avoid using acronyms when stating the study’s objective.

Methods

Did the sampling strategy of the NHSS include any sampling criteria (e.g., sex, age)?

More details on the study sample and characteristics are needed in this section.

How can the authors calculate the incidence based on the NHSS? From my perspective, the authors could assess the prevalence, but not the incidence based on this data. As a result, the regression analyses performed should be estimating prevalence ratios, not incidences.

Results

Table 2: Please include the p-values, and the N besides the percentages.

Please provide confidence intervals also in the text, alongside odds ratios.

Author Response

Reviewer 1:
Overall, I found the study to be well-designed and the methods were described clearly,
for the most part. Please find my specific comments below:
Introduction
Overall, this section nicely frames the study topic; however, I find that there are too
many acronyms to keep track off. In addition, please avoid using acronyms when
stating the study’s objective.
Answer: Thanks for the suggestion. It is true as you commented that there were many
acronyms in this section. We defined every acronyms in the text at first use, and we
provided a list of abbreviations at the end of our manuscript (Line 422) for readers to
keep track. According to your suggestions, we changed those acronyms to their full
names in the statement about the study’s objective in this Introduction section. Please
check it in sentences from Line 105 to 110 (“This study used data from the
Sixth...China’s healthcare insurance.”).
Methods
Did the sampling strategy of the NHSS include any sampling criteria (e.g., sex,
age)?More details on the study sample and characteristics are needed in this section.
How can the authors calculate the incidence based on the NHSS? From my
perspective, the authors could assess the prevalence, but not the incidence based on
this data. As a result, the regression analyses performed should be estimating
prevalence ratios, not incidences.
Answer: Thank you for your attention and suggestions on this part. I’d like to explain
that the NHSS is a kind of national cross-section sample survey, the target population
includes all national population among 31 provinces in China. The overall sampling
strategy of NHSS is economical and effective with a multistage and stratified cluster
sampling method based on division of administrative area. Specific sampling step
was introduced in Data source section (from Line 120 to 124).
The respondents of the survey are the residents among the sampled households.
The residents refer to all registered and non-registered people who have lived in the
household within the past six months. It also includes babies born less than six
months old, newly-married spouses, old people supported in turn, and primary and
middle school students, but does not include non-family members such as nanny.
According to your suggestion about the study sample, we added the overall sampling
strategy, specification of survey respondents and general situation of Jiangsu Province
in the first paragraph (Line 118-120, Line 124-134), and characteristics of sampling
households were added in the end of this section (Line 147-151).
We quite agree with your idea that the difference between incidence and
prevalence should be noticed. As we know, in epidemiology, ‘incidence’ refers to the
frequency of new cases of a disease in a certain range of people within a certain
period of time, while ‘prevalence’ refers to proportion of new and old cases of a
disease in the total population at a given time. To our knowledge, however, in health
economics and social science, there seems no explicit guidance or definitions for us to
distinguish them. Internationally, there are quite a number of literature using
incidence to estimate the occurrence of CHE. For example, catastrophic health care
expenditure in Myanmar: policy implications in leading progress towards universal
health coverage (Chaw-Yin Myint, et al. 2019), Porous safety net: catastrophic health
expenditure and its determinants among insured households in Togo (Esso-Hanam
Atake, et al. 2018), Catastrophic expenditure due to out-ofpocket health payments and
its determinants in Colombian households (Jeannette Liliana Amaya-Lara. 2016),
Catastrophic health expenditure in an urban city: seven years after universal coverage
policy in Thailand (Jain Weraphong, et al. 2013), Progress on catastrophic health
spending in 133 countries: a retrospective observational study (Adam Wagstaf, et al.
2018), Influencing factors of catastrophic health expenditure--analysis based on
CHARLS data (Wenjuan XU, et al. 2018), The drivers of catastrophic expenditure:
outpatient services, hospitalization or medicines?(Priyanka Saksena, Ke Xu,
Varatharajan Durairaj. 2010), The impact of out-of-pocket health expenditure on
household impoverishment: Evidence from Morocco (Meriem Oudmane,et al. 2019).
Many of these studies above are based on cross-section data. Only a few studies
used the concept of prevalence such as the article written by Dilek Basar, et al. (2012):
out-of-Pocket Health Care Expenditure in Turkey: Analysis of the Household Budget
Surveys 2002-2008. It analyses the prevalence of catastrophic out-of-pocket health
expenditure in Turkey. Whereas, the data they analyzed was the Turkish Household
Budget Surveys from 2002 to 2008, which is time-series of repeated cross sections
pooling over seven years. Therefore, in our opinion, we calculated the occurrence of
CHE by “incidence” based on a cross section data is appropriate and consistent with
most literatures.
Results
Table 2: Please include the p-values, and the N besides the percentages.
Please provide confidence intervals also in the text, alongside odds ratios.
Answer: Thank you for your proposals. We have added the P-value and the N besides
the percentages and the confidence intervals alongside odds ratios. Please check the
revision in Table 2.
We feel great thanks for your professional review work on our article.

Reviewer 2 Report

The authors conducted an extensive evaluation of the Critical Illness Insurance (CII) on the catastrophic health expenditure incidence and intensity.   I have some questions on the method and results of this study. Please see comments below.

Please give clear definitions of household size and its potential makeup of a Chinese family in urban and rural areas respectively. It will provide some background information to international readers regarding the size and senior citizens in a Chinese household. 

The makeup of a household (household members >65) and the size of a household may have a combined effect on CHE intensity. In lines 249-250,  “The presence of household members >65 years old was a significant contributor to CHE intensity. In addition, the association between household size and CHE intensity was significantly positive,”   Also, In Lines310-313, “These findings indicate that larger families protect against CHE incidence but not intensity. Although larger families tend to have a good economic ability to reduce the risk of CHE, they are more likely to include more elderly members or children who are susceptible to illness and incur higher costs.”   I suspect that there may have been a multicollinearity relationship between variables household members >65 years old and household size in the model. 

Overall, CII reduced the risk of CHE incidence for the study respondents. (OR=.742) Conversely, higher number of household members having CII increase the odds of CHE.   Can the authors examine the impact of different disease states in a study household on the above two results if you have the data?  Lines 320-327, “A possible explanation is that having CII actually improves households’ capacity to resist economic risk due to critical diseases, and therefore significantly decreased CHE incidence. However, CII implementation also induced residents’ demands for medical services [23], which could lead to higher healthcare expenses, even including uncovered medical expenses that are not on the CII list. Therefore, once CHE occurs in a household, the intensity will be greater, especially in lower income households. As the number of the insured by CII in a household increased, the association of CII with incidence became insignificant. Instead, households with more CII-covered members showed decreased CHE intensity.”

The authors’ argument may be true.  I think one of the possible reasons for the impact of the number of the insured CII may be due to the disease states that CII patients may have.  More CII patients in a household may have higher chance of having expensive diseases in that household. Also, the authors should assess the interaction effect of household members >65 and number of household members with CII on CHE intensity.

I am baffled by the following three conflicting results regarding the impact of registered type (urban vs. rural) on incidence of CHE. Please explain. Line 208 Using a widely used threshold of 40%, CHE incidence rates among urban and rural households were 26.54% and 32.28%, respectively, and the intensity indicates that urban and rural households on average spent 6.46% (overshoot) and 8.44% (overshoot), respectively, over the 40% catastrophic threshold. Line 223 Living in rural areas is associated with a lower likelihood of CHE. Line 275 The CHE incidence and intensity of rural households were significantly higher than those of urban households, irrespective of the threshold used.

Please explain the negative value (-0.057) of difference-Headcount in table 3

Line 278 “Moreover, we found the CHE of rural households was>66.15% of their non-subsistence expenditure. These findings indicate that households, especially in rural areas, are still vulnerable to 9 high medical care costs. Lower income level and the absence of social security in rural areas are among the possible causes. Thus, the government should implement more measures focused on rural households to improve social security system and reduce their CHE.“ However, based on the results in table 1, CHE of urban households was>64.32% of their non-subsistence expenditure.  That is only a small difference of CHE between urban and rural households

Author Response

Reviewer 2:
The authors conducted an extensive evaluation of the Critical Illness Insurance (CII)
on the catastrophic health expenditure incidence and intensity. I have some questions
on the method and results of this study. Please see comments below.
1. Please give clear definitions of household size and its potential makeup of a
Chinese family in urban and rural areas respectively. It will provide some background
information to international readers regarding the size and senior citizens in a Chinese
household.
Answer:Thank you for bringing this to our attention. We have added the definition of
household size in table 1 (in the “Description” column). Household size means
number of persons living together in one household. In China, the proportion of
households with more residents is higher in rural areas than that in urban areas.
Nuclear family, consisting of a married couple and their unmarried children, is the
mainstream form of family structure in urban areas. While in rural areas, there are
more households with 3 or 4 generations living together, which are called stem family
or joint family. And generally, the proportion of the elderly in rural households is
higher than that in urban areas because the majority of the elderly are employed in
agriculture and they are used to living in rural areas. The young and middle-aged tend
to go to urban areas for better employment and live there. These contents about
potential makeup of Chinese family in urban and rural areas have been added in the
fourth paragraph of Discussion section in red colour, from Line 348 to 355.
2. The makeup of a household (household members >65) and the size of a household
may have a combined effect on CHE intensity. In lines 249-250, “The presence of
household members >65 years old was a significant contributor to CHE intensity. In
addition, the association between household size and CHE intensity was significantly
positive,” Also, In Lines310-313, “These findings indicate that larger families protect
against CHE incidence but not intensity. Although larger families tend to have a good
economic ability to reduce the risk of CHE, they are more likely to include more
elderly members or children who are susceptible to illness and incur higher costs.”
I suspect that there may have been a multicollinearity relationship between variables
household members >65 years old and household size in the model.
Answer: Thank you for pointing this out. It is reasonable to suspect a multicollinearity
relationship between variables household members >65 years old and household size
in the model. Our study was based on households units and household size was one of
essential variables in the model. According to your suggestions, we examine the
potential multicollinearity relationship in the model with SPSS software version 24.0.
The result is shown below:
Coefficients
Model
Collinearity Statistics
Tolerance VIF
Household members>65 years old 0.799 1.251
household size 0.494 2.024
Registered type 0.654 1.529
education status of household head 0.784 1.275
Household CII status 0.300 3.333
Number of household members having CII 0.234 4.277
marital status of household head 0.899 1.113
employment status of household head 0.777 1.288
Household members with ≥2 chronic illness 0.894 1.118
Inpatient utilization status of household members 0.131 7.635
Outpatient utilization status of household members 0.642 1.557
Household income level 0.630 1.587
Inpatient service in high-level hospital 0.131 7.658
Outpatient service in high-level hospital 0.639 1.564
Tolerances and variance inflation factor (VIF) methods are commonly used in
the literature to check the multicollinearity. Montgomery D.C. (1991) and O’Brien
(2007) suggested that a Variance Inflation Factor (VIF) greater than 10 are a sign of
multicollinearity. The results in the table show that the lowest tolerance is 0.131 and
the highest VIF value is 7.658. All these values satisfy the critical values (tolerance
<0.1 or VIF >10), which indicate that the extent of multicollinearity among these
factors is allowable.
3. Overall, CII reduced the risk of CHE incidence for the study respondents.
(OR=.742) Conversely, higher number of household members having CII increase the
odds of CHE. Can the authors examine the impact of different disease states in a
study household on the above two results if you have the data?
Lines 320-327, “A possible explanation is that having CII actually improves
households’ capacity to resist economic risk due to critical diseases, and therefore
significantly decreased CHE incidence. However, CII implementation also induced
residents’ demands for medical services [23], which could lead to higher healthcare
expenses, even including uncovered medical expenses that are not on the CII list.
Therefore, once CHE occurs in a household, the intensity will be greater, especially in
lower income households. As the number of the insured by CII in a household
increased, the association of CII with incidence became insignificant. Instead,
households with more CII-covered members showed decreased CHE intensity.”
The authors’ argument may be true. I think one of the possible reasons for the
impact of the number of the insured CII may be due to the disease states that CII
patients may have. More CII patients in a household may have higher chance of
having expensive diseases in that household. Also, the authors should assess the
interaction effect of household members >65 and number of household members with
CII on CHE intensity.
Answer: Thank you for your useful suggestion. We totally agree with your idea that it
is of great necessity and meaning to examine the impact of different diseases status of
households members on the incidence and intensity of CHE. However, it is with
regret that the data related to the specific diseases status of household members is not
available in the database of NHSS. Thanks to your suggestion, we will focus on this
question in our future research.
Considering your advice about the interaction effect of household members >65
and number of household members with CII on CHE intensity. We added an
interaction variable (Household members >65 years old and Number of household
members having CII) in the multiple linear regression model, and also the logistic
regression model, to analyze the interaction effect on CHE intensity and incidence.
The inclusion of a new variable will cause changes of the coefficients of other
variables in the regression model. We modified the Table 4 and Table 5, as well as the
related words in the text according to the new results. Compared to the previous, the
new results of linear regression show that there are only slight changes in values of
coefficients, SD and 95%CI. The coefficient of the interaction variable of household
members >65 years old and number of household members covered by CII on
intensity is -0.006 (P<0.05), which indicates that, compared to households without
members >65 years old, the number of members covered by CII in households with
members >65 years old have more strongly negative effect on the intensity of CHE.
With regard to the result of logistic regression, we found the positive effect of
household members >65 years old on the incidence of CHE is not significant, which
is different from the result of logistic regression without interaction variable. We have
re-written relevant results in the text. Besides, the interaction variable shows a
significantly positive effect on CHE incidence. Households with >65-year-old
members are more likely to have CHE when the number of CII-insured members in
the household increases.
The possible explanation is that most of the members insured by CII in a
household may be the elderly. They are easily suffered from diseases and are
vulnerable to higher healthcare expenses and it does make sense that more CII
patients in a household may have higher chance of having expensive diseases in that
household. However, the CII may provide protection for households having
members >65 years old who are also the insured of CII from high intensity of CHE.
However, the effect of number of CII-covered household members on CHE intensity
keeps negative whatever its interactive correlation with >65-year-old household
members is considered or not. All relevant contents have been added or re-written in
the manuscript. Please check them in Table 4, 5 and the Results (Line 240-244 and
Line 272-273) and Discussion section in red (the 3rd paragraph). We feel great
thanks for your professional review work on our article.
4. I am baffled by the following three conflicting results regarding the impact of
registered type (urban vs. rural) on incidence of CHE. Please explain. Line 208 Using
a widely used threshold of 40%, CHE incidence rates among urban and rural
households were 26.54% and 32.28%, respectively, and the intensity indicates that
urban and rural households on average spent 6.46% (overshoot) and 8.44%
(overshoot), respectively, over the 40% catastrophic threshold. Line 223 Living in
rural areas is associated with a lower likelihood of CHE. Line 275 The CHE incidence
and intensity of rural households were significantly higher than those of urban
households, irrespective of the threshold used.
Answer:Thank you for your careful review. The sentence in Line 299 (namely, Line
275 in your comment, “The CHE incidence and intensity of rural households were
significantly higher than those of urban households, irrespective of the threshold
used.”) in the discussion section is the result of Table 3. We have made it clearer with
adding “In Table 3” at the beginning of the sentence.
With regard to the conflicting results. In table 4, we found a nonsignificant
relationship between registered type (rural or urban) and CHE incidence when we
added an interaction variable (Household members >65 years old and Number of
household members having CII). Therefore, the correlation between registered type
and CHE incidence does not exist. Based on this result, we deleted the sentence
“Living in rural areas is associated with a lower likelihood of CHE” and revised the
related part in discussion section (line 335-336).
5. Please explain the negative value (-0.057) of difference-Headcount in table 3
Answer: Thank you for pointing this out. We are very sorry for our mistakes in this
Table. The value in the Difference-Headcount and Difference-Overshoot rows should
be the absolute difference value between urban and rural CHE incidence (Headcount)
and intensity (Overshoot). Due to our carelessness, there was some calculation error
in table 3. We have checked the result of T-test and made some corrections.
6. Line 278 “Moreover, we found the CHE of rural households was>66.15% of their
non-subsistence expenditure. These findings indicate that households, especially in
rural areas, are still vulnerable to 9 high medical care costs. Lower income level and
the absence of social security in rural areas are among the possible causes. Thus, the
government should implement more measures focused on rural households to improve
social security system and reduce their CHE.“ However, based on the results in table
1, CHE of urban households was>64.32% of their non-subsistence expenditure.
That is only a small difference of CHE between urban and rural households.
Answer: Thank you for your careful review. The sentence “Moreover, we found the
CHE of rural households was>66.15% of their non-subsistence expenditure.” is about
MPO value of rural household in Table 3. It is really true as you commented that
based on the MPO value, the following discussion (“These findings indicate that ...
social security system and reduce their CHE.”) was inappropriate because we didn’t
analyze the difference of MPO between urban and rural households.
Actually, we meant to aim at comparing incidence (Headcount) and intensity
(Overshoot) between rural and urban areas in Table 3 to make the discussion, and
draw the conclusion that incidence (Headcount) and intensity (Overshoot) of rural
households were significantly higher than that of urban households. According to your
comments and our results, we have revised this part. Please see the revision in red
color from Line 303 to 306.
We feel great thanks for your professional review work on our article.

Reviewer 3 Report

The authors state that the purpose of their manuscript is "analyze the extent of CHE among Chinese households and explore the effect of CII and other associated factors on CHE". To achieve this objective the authors use data were derived from the Sixth National Health Service Survey (NHSS, 2018) in Jiangsu Province. The incidence and intensity of CHE were calculated with a sample of 3660 households in urban and rural areas in Jiangsu Province, China. The results found by the authors suggest that: "The proportion of households with no one insured by CII was 50.08% (1833). At each given threshold from 20% to 60%, the incidence and intensity were higher in rural households than in urban ones. CII implementation reduced the incidence of CHE but increased the intensity of CHE. The authors conclude that" Policy effort should further focus on appropriate adjustments such as dynamization of CII lists, medical cost control, increasing the CII coverage rate, and improving the reimbursement level to achieve the ultimate aim of using CII to protect Chinese households against financial risk caused by illness. However, in view of the absence of a theoretical argument that assumes that the authors are making a relevant contribution to the literature, by offering empirical evidence using, authors should invest more effort in the development of work.    I have few points to make:   1. The authors  speech address a specific countrie, without arguing the reasons why they chose this market. Why study Jiangsu Province and not any other region? What is the reason for studying this particular period (September 2018)? And not another period?    2) Regarding the methodology, the Data used for analysis obtained seems appropriate. I think there is a relevant bias in the sample selection quando afirma: "The NHSS of Jiangsu Province was also conducted in September 2018, and the household sample size was 3660, containing a number of 11,550 respondents." Was the number of respondents really this?    3) I suggest that you add in the article a final conclusion of the study highlighting:    - What do we already know about this topic?   - How does your research contribute to the field?    - What are your research's implications toward theory, practice, or policy?   - What would the world lose if this article were not published?  

Author Response

Reviewer 3:
The authors state that the purpose of their manuscript is "analyze the extent of CHE
among Chinese households and explore the effect of CII and other associated factors
on CHE". To achieve this objective the authors use data were derived from the Sixth
National Health Service Survey (NHSS, 2018) in Jiangsu Province. The incidence and
intensity of CHE were calculated with a sample of 3660 households in urban and rural
areas in Jiangsu Province, China. The results found by the authors suggest that: "The
proportion of households with no one insured by CII was 50.08% (1833). At each
given threshold from 20% to 60%, the incidence and intensity were higher in rural
households than in urban ones. CII implementation reduced the incidence of CHE but
increased the intensity of CHE. The authors conclude that" Policy effort should
further focus on appropriate adjustments such as dynamization of CII lists, medical
cost control, increasing the CII coverage rate, and improving the reimbursement level
to achieve the ultimate aim of using CII to protect Chinese households against
financial risk caused by illness. However, in view of the absence of a theoretical
argument that assumes that the authors are making a relevant contribution to the
literature, by offering empirical evidence using, authors should invest more effort in
the development of work.
I have few points to make:
1. The authors speech address a specific countrie, without arguing the reasons why
they chose this market. Why study Jiangsu Province and not any other region? What
is the reason for studying this particular period (September 2018)? And not another
period?
Answer: Thank you for pointing this out. Firstly, we chose this specific country
because we are familiar with it. Besides, China has had almost nationwide coverage
of basic health insurance system but the service provisions and financial protection of
the system in China are far from sufficient. China launched critical illness insurance
in 2012 and realized fully implementation of CII in 2016, which aimed to protect
urban and rural residents against large medical expenses and relieve medical
expenditure burden. China, as a developing and large population country, plays an
important role in achieving universal health coverage and accelerating the pace of
achieving the sustainable development goals. It is necessary to evaluate the
implementation of Chinese health policy and provide some significant empirical
evidence for reforming China’s healthcare insurance and some experience reference
for other developing countries.
We studied Jiangsu Province because it is one of the pilot provinces of national
health reform. It will be meaningful to study the implementation status of CII, which
is one of the measures of Chinese national health insurance system reform.
Additionally, some of our authors had participated in the survey and we only had
access to data of Jiangsu Province from the sixth NHSS (2018) database. We have
added some background information about Jiangsu Province in the manuscript. Please
see in the first paragraph of Data source section, from Line 129 to 132.
The National Health Service Survey was organized by Chinese National Health
Commission every 5 years since 1993, and the Provincial Commission of Health is
responsible for the specific survey in their region. China usually makes an overall
plan for the reform of the national health system every five years and the survey may
provide data support for it. The first, second, third, fourth and fifth NHSS was
conducted in 1993, 1998, 2003, 2008 and 2013, respectively. Regularly, the sixth
NHSS was in 2018. The sixth NHSS of Jiangsu Province was conducted in September,
which was stipulated by Jiangsu Commission of Health.
2. Regarding the methodology, the Data used for analysis obtained seems appropriate.
I think there is a relevant bias in the sample selection quando afirma: "The NHSS of
Jiangsu Province was also conducted in September 2018, and the household sample
size was 3660, containing a number of 11,550 respondents."Was the number of
respondents really this?
Answer: Thank you for pointing this out. The respondents was 11550. Because of the
household unit rather than individual for our study analysis, the final households
sample was 3660 after the elimination of missing individual value. We have re-written
this part to more accurately account for it. Please see in Data source section, from
Line 133 to 135.
3. I suggest that you add in the article a final conclusion of the study highlighting:
What do we already know about this topic? How does your research contribute to the
field? What are your research's implications toward theory, practice, or policy?
What would the world lose if this article were not published?
Answer: Thank you for your suggestion. We have re-written Conclusion section
according to the suggestion. The revision as follows:
To the best of our knowledge, some theoretical studies researching the
implementation effect of CII in China mainly focused on describing and comparing
the design and development of the program policy. Due to a short period time since
broad CII implementation and lack of empirical data, few existing studies provide
strong empirical evidence on the effect of CII on CHE. Our findings, basing on
Chinese latest and authoritative data of representative region, not only reveal that
urban-rural discrepancies in the incidence and intensity of CHE but also show the
mixed effect of CII and some other associated factors on CHE. More government
measures focusing on rural households should be implemented to reduce their
catastrophic medical expenses. Policy adjustments such as a dynamic CII list, medical
cost control mechanisms, broader CII coverage, or improved reimbursement levels
should be developed to provide solid protection for Chinese households against
financial medical risk. Additionally, other socioeconomic factors significantly
affecting CHE indicate that policies aimed at reducing CHE should address the
socioeconomic factors of healthcare outcomes among urban and rural households.
Our study clarifies the implementation effect of its CII on financial risk protection in
Jiangsu Province and it also provides reference and guidance for reforming China’s
healthcare insurance.
We feel great thanks for your professional review work on our article.

Round 2

Reviewer 1 Report

The authors have addressed all my comments and suggestions. I think that the manuscript has greatly improved and that it is suitable for publication in its current form.